# The Sociocultural Influences on Breast Cancer Screening among Rural African Women in South Africa

**DOI:** 10.3390/ijerph20217005

**Published:** 2023-11-01

**Authors:** Nelisha Sarmah, Maureen Nokuthula Sibiya, Thandokuhle Emmanuel Khoza

**Affiliations:** 1Faculty of Health Sciences, Durban University of Technology, 7 Ritson Rd, Durban 4001, South Africa; thandokuhlek@dut.ac.za; 2Division of Research, Innovation and Engagement, Umlazi Campus, Mangosuthu University of Technology, Umlazi 4031, South Africa; sibiya.nokuthula@mut.ac.za

**Keywords:** breast cancer, culture, belief, rural, South Africa

## Abstract

The incidence of breast cancer in South Africa is increasing, with rural South African women presenting with advanced stages of the disease. A woman’s breasts are a symbol of her womanhood; they also constitute a social definition of her femininity. Women with breast cancer in rural South Africa are heavily stigmatized and suffer from various sociocultural interpretations of the disease. Breast cancer is frequently interpreted in rural South Africa as a symbol of witchcraft, sin, and punishment, and traditionally, it is treated by offering animal sacrifices, consumption of herbs, and prayer to ancestors. Using care-seeking behaviour theory as the theoretical framework, we intend to explore the sociocultural factors influencing breast cancer screening practices among rural South African women. A qualitative exploratory study was conducted using semi-structured interviews with 22 rural South African women selected by purposive sampling. Thematic analysis was used to analyse the data. The study identified four sociocultural factors influencing women’s practices of breast cancer screening in rural South Africa, including psychological factors, habits, beliefs, and perceptions of healthcare. Women in rural South African communities have deep-rooted traditional beliefs and practices regarding breast cancer. Consequently, this influences women’s preventative health behaviours regarding breast cancer screening. The development of culturally appropriate health education programs involving traditional healers and influential community leaders is essential to increasing the number of women being screened for breast cancer in rural South Africa.

## 1. Introduction

Breast cancer is not only the most diagnosed cancers worldwide, but it is also the leading cause of cancer death in over 100 countries [1]. The disease is also the most common cancer among women in South Africa, accounting for 22.6% of all female cancers and 16% of cancer deaths [2]. The majority of women diagnosed at a tertiary hospital in South Africa are predominantly African women from rural areas and are three times more likely to be diagnosed late than women living in urban areas [3]. Breast cancer affects women of all races in South Africa, with a lifetime risk of one in 25 [4]. According to previous South African studies, 63.4% of breast cancer patients of African descent are in stages III or IV of the disease [5]. The most common reason for consultation and findings at clinical examination was a breast mass, with 89.6% of tumours being invasive ductal carcinomas and 3.7% being invasive lobular carcinomas [5]. Rural women in South Africa have been strongly associated with late presentation of breast cancer due to a high unemployment rate, lack of resources, distance to the nearest health facility, lack of transportation services, lack of education, insufficient knowledge and awareness, and lack of family support [6]. Furthermore, the delay in seeking medical assistance at a hospital was over two months because rural South African women sought medical assistance from traditional healers. Consequently, most cancer patients in rural areas present at a late stage of the disease.

The burden of cancer is disproportionately felt in low- and middle-income countries (LMICs), where the majority of new cancer cases and 70% of cancer deaths occur [7]. The lack of domestic and donor funding has resulted in poor access to cancer prevention, screening, and diagnostic services as well as low levels of access to quality cancer treatment. Inequalities in access to cancer prevention, screening, diagnosis, and treatment play a significant role in the high rates of cancer mortality in South Africa [7]. As part of cancer prevention and control, the World Health Organization (WHO) recommends screening and early detection as the two most important components for countries with a high prevalence and mortality rate of breast cancer [8]. The importance of early detection methods to improve treatment options for women has been highlighted in numerous studies worldwide [2,3]. Breast cancer can be detected using mammography and breast self-examination (BSE) [9,10]. Although mammography is the mainstay for early detection of breast cancer, it is not accessible to millions of women in developing countries because it is financially and technically challenging to implement and sustain because it requires well-trained radiographers and radiologists, and investment in pathology and treatment centres [11,12]. Due to a lack of resources, mammography screening is not readily available in rural South Africa. Most rural areas in South Africa are located a considerable distance from the nearest healthcare facility offering mammography screening. This results in significant travel costs as most rural residents are of low socioeconomic backgrounds. Despite controversy concerning the indications for BSE, recommendations differ among screening task forces, medical academies, advocacy groups, and regional branches of the World Health Organization. Even though the South African Department of Health (DoH) introduced the Breast Cancer Prevention and Control Policy in 2017, it lacked the resources to implement and sustain a national screening program [13]. For this reason, the South African DoH recommends clinical breast examinations and BSE as early detection methods for breast cancer. Despite its safety, ease of access, and high recommendation for women in low-resource settings, BSE is not widely adopted by South African women living in rural areas.

Globally, low breast cancer screening rates can be attributed to a lack of knowledge, a lack of awareness, cultural influences, and socioeconomic factors. Women’s knowledge of breast cancer screening varies across nations probably due to their cultural beliefs and socio-economic levels [14]. The current literature indicates that approximately 27 million people in South Africa, especially Africans, depend on traditional medicine for primary healthcare needs [15]. Traditional healing practices form part of South African culture and spiritual life [15]. Cultural beliefs influence how individuals interact with the world and behave under certain circumstances. This may be seen as a combination of religious beliefs, socially accepted norms, and traditions. Research has shown that personal philosophies are directly influenced by cultural values and belief systems, which are often reflected in their health-seeking behaviours [7]. There is a growing body of research that suggests cultural barriers influence breast cancer screening and overall healthcare [14]. Due to the scarcity of research, this paper explores the sociocultural factors influencing breast cancer screening from a rural South African perspective, where most women present with advanced stages of breast cancer. Furthermore, this publication provides valuable information that can be used to develop culturally and socially appropriate breast cancer prevention programs and campaigns among rural South African women.

## 2. Theoretical Framework

This study utilized the care-seeking behaviour (CSB) theory as a theoretical framework. This theory was originally developed to explain and predict preventative health behaviours [16]. The CSB theory includes a wider range of constructs than any other health behaviour theory, allowing for further research. Furthermore, the CSB theory posits that an individual’s potential to engage in health behaviours is influenced by multiple factors, including psycho-social, clinical, socio-demographic, and facilitating factors [17]. It has been proposed that the CSB theory can be utilized to explain why people participate in health promotional programs such as mammography screening by accounting for external factors which could negatively affect screening behaviour [18]. According to the theoretical framework, self-efficacy impacts the relationship between perceived benefit and behavioural adoption of breast screening practices as well as the relationship between subjective norms and breast screening practices. In line with this finding, a study conducted in the United States revealed that participants’ beliefs, affects, cultural norms, and self-efficacy were strongly related to their behaviour with regard to breast cancer screening [19]. Using the CSB theory, a study was conducted among Hmong women to describe their beliefs, feelings, norms, and external conditions regarding breast and cervical cancer [20]. The findings indicate that women’s beliefs, lack of knowledge, feelings of embarrassment, and cultural norms influence their behaviour regarding breast cancer screening. The use of the CSB theory is further suggested as a means of guiding public health interventions aimed at improving culturally sensitive screening for women [20]. Several psycho-social constructs of the CSB theory such as affect, belief, norm, and habit, together with clinical and sociodemographic factors indirectly influence CSB [18]. In this paper, psycho-social constructs, namely affect, belief, habit, and norm, are explored to identify the socio-cultural factors influencing breast cancer screening among rural South African women:

Affect: refers to a feeling of anxiety that may result from embarrassment or a diagnosis and these affectual concerns may lead to individuals refraining from performing preventative health behaviours [21,22].

Habit: refers to the way one acts when one is experiencing symptoms (for instance, whether or not one seeks medical attention promptly) [17].

Belief: reflects the overall value or importance of CSB [17]. Most studies indicate that women’s beliefs are heavily influenced by cultural factors [7].

Norm: consists of the perception of morally acceptable behaviour towards seeking health care [22]. The CSB theory asserts that norms include social norms, which reflect the beliefs of others regarding seeking health care; personal norms, which represent one’s views about seeking medical attention; and interpersonal norms, which reflect an agreement between individuals to seek medical attention [22].

## 3. Materials and Methods

### 3.1. Study Design and Setting

This paper presents the findings of a qualitative study conducted between August and October 2022. This study examines the sociocultural factors that influence the practice of breast cancer screening among rural South African women living in the iLembe District of KwaZulu-Natal (KZN). The district is divided into four local municipalities (areas): Mandeni, KwaDukuza, Ndwedwe, and Maphumulo [23]. We chose the iLembe district as the study location since it is primarily rural with a wide distribution of clinics. In this district, 52% of the population is female [23]. In some cases, patients are referred to tertiary hospitals located far away. This is because these clinics offer limited services to communities, resulting in high travel costs [23]. In this district, four clinics located in four different areas were used for recruitment and interviewing participants. Several clinics were selected for this study as they serve a majority of the sample population and are easily accessible to the participants. Moreover, the researchers could not identify any other venue within rural areas that would attract many rural South African women. The local church in some areas may be a distance away and will not attract as many people as the local clinic within that region.

### 3.2. Study Participants and Sample

The study sample included South African women living in the iLembe District of KZN. Participants were recruited from four clinics in the district located in different areas. Participants within these clinics were presented with a research information sheet and asked to consent to participate in the study. This study employed a purposive sampling technique to select participants who were South Africans of African descent, living in the rural iLembe district in KZN province, aged 20 to 60, and who consented to participate in the study. Following the explanation of the research information sheet and the successful recruitment of participants, a written consent was obtained, and then the interview process commenced. This study included 22 participants as a result of data saturation.

### 3.3. Data Collection

Data were collected at four healthcare facilities (clinics) in the iLembe district of KZN between August 2022 and October 2022. It was found that these healthcare facilities served a large sample of the population. Therefore, it was the most suitable option. The duration of each interview ranged from 30 to 40 min. Interviews were conducted in a designated area within the clinic to ensure confidentiality. Interviews were carried out in English, however a translator was available for participants to speak in their native language of isiZulu if they desired. To gain a deeper understanding of the sociocultural factors influencing breast cancer screening among South African rural women, a semi-structured, one-on-one interview approach was employed. Please refer to Appendix A for the interview guide. The use of a standardized interview guide assured consistency, stability, and repeatability of participants’ perspectives. This ensured the credibility of the interviews. As a result, the researchers established consistency in the information and verified the traditional beliefs, practices, and perceptions of rural South African women. Once data saturation had been reached, the interviewing process was terminated. Saturation of data occurs when no new information is provided by the interviewees.

### 3.4. Data Analysis

Transcripts of the interviews were transcribed verbatim by one of the researchers, with participant names replaced with codes to ensure confidentiality. Thematic analysis was chosen as the method of data analysis. The purpose of this was to provide an in-depth overview and description of the data comprehensively and interpret various aspects of the research topic. Thematic analysis consists of six stages [24]:The researchers read the interview transcripts repeatedly to identify meanings and patterns, as well as to make preliminary observations.Upon reading and understanding the transcripts, initial codes are developed and written. From the codes, the researchers develop a series of categories and subcategories from which themes are derived. The process is repeated continuously to ensure that all possible codes have been recorded.Once all codes have been identified across the data set, the researchers identify overarching themes. Some of these codes are subsequently classified as primary themes, others as sub-themes, and some are discarded.Based on the supporting data, the researchers further categorize or merge the themes.The themes and supporting data from the interviews are then summarized.In light of the study’s data extracts, the researchers draw conclusions that confirm the prevalence of the theme.

As suggested by Lincoln and Guba, this study was evaluated according to four criteria for establishing the trustworthiness of qualitative research: credibility, dependability, confirmability, and transferability. A high level of credibility was achieved as a result of extensive engagement with the participants. A record of the raw data collected during each interview was kept for future reference to ensure reliability. An audit trail detailing data collection, analysis, and interpretation was used to establish confirmability. The transferability of this study was achieved by providing detailed descriptions of the research setting and methods, thus confirming the authenticity and validity of the study. This allows for future research to be based on the findings.

### 3.5. Ethics

Ethical clearance was obtained from the Institutional Research Ethics Committee (IREC 157/22) at the Durban University of Technology for this study. The gatekeeper’s permission was obtained from the KZN Department of Health and the iLembe District Manager. A consent form was signed by all participants to participate in this study and to permit the researchers to record the interviews. All responses were kept anonymous and confidential. During the interview, the participants’ names were not used, but a number was assigned to each based on their order of interview. P1 was the first person interviewed, followed by P2 up to P22. Data were stored electronically and password protected.

## 4. Results

### 4.1. Demographics of Participants

Table 1 presents the demographics of the study participants. Participants ranged in age from 20 to 65 years old, with the majority possessing a high school education. It is widely recognized that extreme poverty severely constrains the level of education in many African countries, which has grave implications for the provision of quality education. As a result, rural areas in South Africa are perceived as under-resourced and marginalized. Therefore, only a small percentage of participants had tertiary education. Additionally, the same number of participants were employed and unemployed, with an average of two children per woman. The majority of participants were unmarried.

### 4.2. Themes

The supporting extracts of the interviews revealed a number of themes related to the theoretical framework (CSB). The data were analysed and themes were formulated by two of the researchers. Four themes were identified as influencing the practice of breast cancer screening among rural South African women. As part of the CSB theory, the construct “affect” is identified as the theme of “psychological influences,” which suggests feelings of fear, stigmatization, social support, and attitude. To better understand this construct, participants were asked to describe how they would feel if they discovered a lump in their breast. Moreover, follow-up questions involved exploring their comfort level with discussing their breast health with family, friends, and community members. The second construct, “habit”, was identified as a theme of “preventative healthcare habits”. In order to understand rural South African women’s healthcare habits, these women were asked about their habits when experiencing medical symptoms. As a result, the interviewer was prompted to explore further the various habits mentioned in the study. Thirdly, the construct of “belief” was identified as a theme of “breast cancer beliefs”. In drafting the questions, consideration was given to the existing literature as well as the theoretical framework of this study. Therefore, participants were asked to describe their traditional or cultural understanding of breast cancer and the treatment options available to them. Lastly, the construct “norm” was applied to the theme of “healthcare perception”. To better understand the perception of healthcare at a community level, participants were asked whether women are encouraged to practice BSE across the community. The findings of each of the four themes mentioned will be presented in the following sections.

#### 4.2.1. Psychological Influences

Breast cancer screening among rural South African women is influenced by four factors: fear, cultural stigmatization, social support, and attitude. When it comes to breast cancer and discovering a lump in the breast, most women express fear.


*“I will be heartbroken. As you can see my breast is small, imagine if I can find the lump. I will be scared.”*
(P8).


*“…. cancer is something that you should be afraid of. It’s not just a disease, it’s very dangerous.”*
(P18).

The fear of breast cancer is often attributed to the belief that the disease is incurable and fatal.


*“This illness is like we scared to have it because we think that it’s not curable, and you’re going to die.”*
(P3).

Additionally, participants frequently referenced the concept of stigmatization in their responses to the interview questions. According to several participants, South African women diagnosed with breast cancer are heavily stigmatized by other Africans. It is therefore less likely that women will engage in breast cancer screening or discuss it with others. The following excerpt illustrates some of the reported stigmatization:


*“They will mistreat her. They will see her as a different person like she’s not qualified to be a woman.”*
(P2).


*“When I say community, I mean women. Ladies! She will be such a shame. To males, no one will ever want to marry her and with her family, she’ll be a disgrace.”*
(P10).

Interestingly, some participants said women with breast cancer or breast cancer symptoms are stigmatized and compared to HIV-positive individuals.


*“When you start losing weight and you look sickly, they go around talking about you, that you’ve contracted AIDS.”*
(P13).

Consequently, this leads to the factor of social support. It was agreed by participants that South African women’s preventative health behaviours are greatly affected by the social support they receive from their families, friends, and communities. The likelihood of women being able to communicate their breast-related concerns is higher when they have a good support network. A positive response was noted by some participants when asked about how easy it is to discuss breast problems with family and friends:


*“It’s easy to talk to them, they understand.”*
(P4).

Additionally, the majority of participants expressed a favourable attitude regarding their willingness to practice BSE and to participate in breast cancer programs in their respective communities.


*“I want to learn more. I want to know more.”*
(P3).


*“Yes, I will go because I will learn more, and I will understand more about breast cancer.”*
(P11).

#### 4.2.2. Preventative Healthcare Habits

It cannot be overstated how important it is to understand women’s healthcare habits, particularly when one is experiencing a medical condition. To gain a better understanding of women’s preventative healthcare behaviours regarding breast cancer screening, we sought to understand habits. The one-on-one interviews conducted with African women revealed that they have three distinct healthcare habits, namely seeking healthcare services, self-treatment options, and traditional practices. Several participants reported that they would consult a healthcare facility if they had a breast-related problem or any other health-related problem, indicating their confidence in the healthcare system.


*“Obviously I will go to the clinic to seek medical help.”*
(P6).

There were, however, several participants who indicated that they would not visit a health facility and would rather practice self-treatment options. Self-treatment options for medical symptoms included non-prescription pain medications found in their local shops, as well as turmeric roots and ginger.


*“I will just make what the people tell me I must make and drink, like ginger, fresh turmeric roots.”*
(P12).


*“If it’s something painful, I get some painkillers and use them and see if the pain goes away.”*
(P20).

Other participants indicated that they would engage in traditional practices based on their religious beliefs. The traditional prayer herbs sought from traditional healers in their communities were consumed in this manner. Furthermore, participants reported that they may need to participate in slaughter ceremonies of animals and pray to their ancestors, as prescribed by traditional healers.


*“I use traditional herbs.”*
(P2).


*“If you get sick, they will consult your late grandfathers….. Maybe they will say, maybe we have to slaughter a cow.”*
(P3).

#### 4.2.3. Breast Cancer Beliefs

Participants’ beliefs regarding breast cancer emerged as a dominant theme during the interview process. Two distinct factors were identified under this theme, namely healthcare beliefs and traditional beliefs. Several participants indicated that they would consult a health facility if they were experiencing breast-related problems or if they had any other medical concerns. It was determined that these participants had demonstrated an understanding of the concept of breast cancer and BSE earlier in the interview process. As a result, they held strong healthcare beliefs.


*“Breast cancer is where you have carcinogens or cancer in your breast or where you have a tumour in your breast.”*
(P13).


*“Healthcare provider first. Because they’re able to run the test, more tests to see what’s wrong with you.”*
(P18).

Additionally, there were a substantial number of participants who held strong religious or traditional beliefs regarding breast cancer. Participants reported several traditional interpretations of breast cancer. Among these were those that believed breast cancer was the result of witchcraft, punishment, or sin.


*“Mostly, they think of witchcraft. They think maybe you are cursed, maybe something you’ve done, something in the past that you are not supposed to do now and God is punishing you, things like that.”*
(P20).

Some women, however, believed that breast cancer signified the ‘cutting’ of a woman’s breast.


*“Cut your breast completely.”*
(P20).

#### 4.2.4. Healthcare Perception

This theme sheds light on individuals’ morally righteous beliefs regarding health behaviour in terms of seeking health care, as well as their willingness to act on those beliefs. There were three factors found to influence health care perception, namely personal norms, social norms, and interpersonal norms. Personal norms explored participants’ perceptions of seeking medical care and how they understand and perceive BSE. In this study, it was revealed that a few participants were confident in practicing BSE. However, other participants were unsure of how to perform this screening procedure correctly. In addition, some participants indicated that they had sought out information about breast health concerns through the use of mass media, such as Google.


*“I don’t know if I’m doing it the correct way. That’s the problem.”*
(P5).


*“I Google and search, and they asked what kind of bra I am wearing. So, I wrote, I wear a bra with the underwire and they say you must stop wearing and try to wear the one that is comfortable with no wire. So, I went for those bras and the pain stops.”*
(P3).

Social norms reflect society’s perception of preventative healthcare and breast cancer screening. It was found that a lack of awareness and knowledge of breast cancer screening was prevalent among family, friends, and the community.


*“They know nothing about it.”*
(P1).

In addition, it was found that traditional beliefs play a significant role in how society interprets breast cancer.


*“They only believe in traditional things. So, it’s not easy to talk to them. And some will just treat you like you’re cursed or something.”*
(P2).

All participants agreed, however, that raising awareness of breast cancer screening among family, friends, and communities would encourage more women to undergo screenings.


*“They will understand if there was someone to explain to them what breast cancer is and how it is cured.”*
(P2).


*“They will understand if there could be someone to explain to them what breast cancer is.”*
(P3).


*“Teach them, have a meeting, give health education it’s the only way we can help them.”*
(P6).

Observations of interpersonal norms revealed the understanding between individuals regarding the necessity of seeking medical attention. The participants reported that women who were encouraged to practice breast cancer screening were more likely to do so.


*“If we have someone who know about it, like for instance, I know about the examination. So, if I teach my family obviously they will practice it.”*
(P1).

## 5. Discussion

This study explores the sociocultural factors that influence breast cancer screening among rural South African women. Furthermore, we explored breast cancer perceptions from a sociocultural perspective at a community level and identified challenges and potential opportunities associated with breast cancer screening in rural South Africa. Based on the research findings presented in this paper, psychological influences, habit, belief, and healthcare perception were identified as sociocultural factors influencing breast cancer screening among rural South African women.

Participants expressed a fear of death due to the misconception that breast cancer is incurable. According to a previous study, women who have been diagnosed with breast cancer feared a mastectomy and death. This prevented them from consulting their physicians or reporting to a health centre [25]. Additionally, some participants reported that their communities fiercely discriminate against women with breast cancer. In some cases, participants expressed concerns that they would feel ‘less of a woman’ if their breast was removed. A woman’s breasts are a symbol of her womanhood; they also constitute a social definition of her femininity. The loss of a breast would suggest that a women is less of a woman and that their femininity has diminished. Researchers in Tanzania found that societal stigmatization of cancer contributed to some participants’ fear of disclosing symptoms to others, resulting in a delay in seeking medical assistance [25]. Furthermore, previous studies have reported that participants expressed fear of deformity after surgery, a belief that cancer was incurable, or a belief that going to the hospital was a death sentence [25]. The findings of the present study are consistent with those of the previous study. As a result, many women do not practice BSE for fear of discovering a breast lump and being socially rejected by their family, friends, and the community at large. Per this study’s findings, the social support received by rural South African women significantly impacts women’s preventative health behaviours. In a Nigerian study, families were found to be a significant source of motivation and encouragement for women to continue to attend screenings [26]. A woman who has a good support network is more likely to be able to communicate her concerns regarding breast cancer.

It has been found in a previous study that women with preventative habits are 4.8 times more likely to adhere to screening recommendations, such as mammograms, than women without such habits [27]. As a result, women who practice preventative habits are twice as likely to undergo screening for breast cancer. According to studies conducted in other African countries, many women prefer herbal medicine and traditional practices to established screening and treatment methods [25]. One study reported that only 29% of women who recognized breast cancer signs and symptoms sought medical attention, whereas 46% opted for traditional treatment [28]. It was noted in this study that although there were participants with strong traditional beliefs, habits, and perceptions, there were also participants with stronger healthcare beliefs and perceptions who sought medical assistance.

Traditional beliefs and practices play a significant role in breast cancer screening among rural South African women. A common interpretation of breast cancer is that it represents witchcraft, sin, and punishment. It is commonly treated by sacrificing animals, consuming herbs, and praying to ancestors. According to a study conducted in Nigeria, breast cancer is the result of evil spirits, curses, or promiscuity and can only be cured spiritually [26]. Even though this study did not identify any positive traditional beliefs promoting breast cancer screening, it did identify the importance of traditional healers within communities. Rural South African women commonly consult traditional healers for the treatment of breast cancer, which has been identified as a negative barrier to breast cancer screening. Due to their high social influence in these communities, there is a clear opportunity to involve traditional healers in breast cancer screening programs to raise awareness of the disease. Traditionally, African women need the permission of their spouses before receiving medical care at a hospital or attending a health education forum [28]. Often, women put their health at the bottom of the priority list when balancing family responsibilities and health. This study could not validate these findings since the majority of participants were unmarried. However, some participants indicated that if diagnosed with breast cancer, the needs of their families would take precedence and they would remain silent. This is due to the embarrassment and stress it may cause.

Several other studies suggest that women who are knowledgeable about breast cancer awareness may not practice breast cancer screening because of social factors that may influence their health-seeking behaviour [29]. In rural communities, for example, negative experiences can be passed around through word-of-mouth, causing individuals to be reluctant to seek medical attention and mistrust Western medicine [17], leading to decreased access to healthcare. Women with strong social norms regarding breast cancer have twice the rates of screening as women with weaker social norms [27]. In terms of screening involvement and presentation, women’s attitudes were influenced by their sources of information. The more knowledgeable a society is about breast cancer, the more likely it is to encourage individuals to seek medical attention.

## 6. Limitations

The present study focused solely on sociocultural factors that influence breast cancer screening in rural South Africa. There may be other factors influencing breast cancer screening in rural South Africa, such as traditional healers’ and tribal leaders’ perceptions of healthcare. Future studies should explore these factors in greater detail. Furthermore, the findings of this study should not be construed as representative of all African women in South Africa. Rather, we present a snapshot of the narratives provided by a small sample of rural South African women, and our analysis reflects their viewpoints. It should be noted that although participants responded to the interview questions in English, their native language is isiZulu. There is a possibility that participants may not have understood the interview questions in their entirety.

## 7. Conclusions

This study suggests that sociocultural factors influence rural South African women’s preventative health behaviours and breast cancer screening practices. Furthermore, it was understood that rural South African women’s willingness to participate in breast cancer screenings was largely dependent on social support, traditional beliefs, and practices. In addition, it was found that South African rural women have a variety of preventative healthcare habits, which may contribute to further delays in seeking medical treatment. Moreover, this study suggests that rural South African women are more likely to present at a later stage due to fear, cultural stigmatization, and embarrassment as a result of a lack of social support from family and friends. Enhancing community engagement and breast health education are essential to improving the detection of breast cancer in rural South African communities. Although the present study acknowledged the role of traditional healers in breast cancer treatment, it did not examine their role in breast cancer screening. Therefore, future research should examine the role of traditional healers in the screening and prevention of breast cancer in rural South African communities. Additionally, there is a need for culturally sensitive breast cancer campaigns, involving key stakeholders, like traditional healers, tribal leaders, school educators, and local health facilities, who are trusted by the community.

## Figures and Tables

**Table 1 ijerph-20-07005-t001:** Demographics of participants.

Participants	Age	Level of Education	Marital Status	Employment Status	Number of Children	Municipal Residence
P1	31–40	H/S	M	FT	None	Ndwedwe
P2	20–30	H/S	S	U/E	1	Ndwedwe
P3	20–30	H/S	S	U/E	3	Ndwedwe
P4	31–40	H/S	S	U/E	3	Ndwedwe
P5	20–30	T/E	S	PT	None	Ndwedwe
P6	31–40	T/E	M	FT	More than 3	Ndwedwe
P7	20–30	H/S	S	U/E	None	Ndwedwe
P8	20–30	T/E	S	FT	1	KwaDukuza
P9	41–50	T/E	W	FT	3	KwaDukuza
P10	31–40	H/S	S	PT	More than 3	KwaDukuza
P11	31–40	H/S	S	FT	1	KwaDukuza
P12	41–50	H/S	S	U/E	None	KwaDukuza
P13	51–65	T/E	S	FT	2	KwaDukuza
P14	41–50	H/S	M	FT	2	Maphumulo
P15	20–30	T/E	S	FT	2	Maphumulo
P16	31–40	H/S	S	PT	2	Maphumulo
P17	20–30	H/S	S	FT	None	Maphumulo
P18	41–50	H/S	S	PT	2	Mandeni
P19	41–50	H/S	M	U/E	More than 3	Mandeni
P20	41–50	H/S	S	U/E	1	Mandeni
P21	31–40	H/S	S	FT	2	Mandeni
P22	20–30	H/S	S	FT	2	Mandeni

Key: P = Participants, H/S = High School, T/E = Tertiary Education, S = Single, M = Married, W = Widowed, FT = Full-time Employed, PT = Part-time Employed, U/E = Unemployed.

## Data Availability

Not applicable.

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
