# Peer review of "The Sociocultural Influences on Breast Cancer Screening among Rural African Women in South Africa"

_ijerph, 2023, doi:10.3390/ijerph20217005_

Round 1
Reviewer 1 Report
Comments and Suggestions for Authors
This is a well-conducted study, socially of high relevance. The paper would attract more readers if it mentioned, in a few words, what is stated in:
Line 206: about the stigmatization of this disease
Line 71: Care-Seeking Behaviour Theory was the theoretical framework.
Lines 364-365: as the social representations of the causes of breast cancer, as well as its native treatments
Lines 388-391: about the importance of the breast to feel a complete woman
Abstract in lines 13-15 can be rewritten to accommodate these recommendations.
Concerning the Method, it would be necessary:
Lines 128-129 – Add the script of the questions (attached) or some of them used in the interview – so other researchers could evaluate better the adequacy of CSB and maybe use it in their future studies.
Reviewer 2 Report
Comments and Suggestions for Authors
The importance of the topic is discussed fluently in the introduction, but the method and discussion section need to be strengthened. My suggestions are marked on the file.

Reviewer 3 Report
Comments and Suggestions for Authors
The paper deals with the important public health issue of breast cancer and specifically the sociocultural influences on breast cancer screening behavior among African women who live in rural areas in S. Africa. It was interesting to see how cultural beliefs still play a major role in the decision making process of breast cancer screening behavior in this part of the world. Here are my comments that could strengthen the quality of the paper.
Introduction;
*Lines 41-42: It is stated that early detection of breast cancer is achieved through breast self examination and mammography. There is no reference to support this claim. In fact, most international guidelines for breast cancer screening, indicate that screening mammography is the primary modality for early detection of breast cancer. Women should be aware of any changes in their breasts, however, Breast Self Examination is not recommended anymore as a tool for early detection of breast cancer. https://www.komen.org/breast-cancer/screening/breast-self-exam/. Now, it is not clear if the women who live in these rural areas in S. Africa have access to screening mammography through clinics, or clinical breast examination. Why is the research centered on BSE?
* Is there any specific epidemiological picture regarding breast cancer among the women who live in the specific area where the study takes place? What are the cancer inequalities in this region?
* Has the CSB theory ever applied successfully in breast cancer screening behaviors and what were the results of those applications?
* What is the research question that this study is trying to answer?
Material and methods:
· What was the target population? What are the criteria of entering the study in terms of breast cancer history? Did they recruit women who were healthy or women who have been diagnosed with breast cancer?
· The sample was very small. Why did the authors recruit the participants from the clinics and not from community centers like the church? If the purpose of the study was to find out how sociocultural factors affect the decision of a woman with average risk for breast cancer to do a mammogram or BSE then it would be more appropriate to recruit women who are asymptomatic and free from breast cancer from a neutral setting. Past or current experience of breast cancer or being interviewed at a medical setting can bias their motivation to do the preventative care.
· How did the theory help in the development of the questions in the interview path? Has the interview guide been pretested before the actual data collection? There is no information about the discussion guide in the manuscript.
· The authors state that a purposive sampling method was used to recruit the participants. How was this done? Was the sample theoretically divided by age, breast cancer experience etc?
· How many people were involved in the analysis? Because qualitative research is very subjective, at least 2 people are needed to identify the themes of the study.
· In line 131 authors state that “triangulation was achieved by conducting interviews by 4 different clinics”. Please provide an example that shows how triangulation was achieved.
· Regarding themes, how did the authors decide that an idea that was mentioned in the interviews was a theme?
Results:
The authors state various themes but it is not clear how the CSB has guided them to derive to those themes. For example, what were the questions asked based on the theoretical concepts that were used to elicit answers and thus the themes? The authors emphasize the importance of traditional beliefs in seeking preventative action and mostly viewed as barriers towards breast cancer screening. Were there any positive traditional beliefs that could promote breast cancer screening?
Discussion:
How does the role of woman in the society might play a role in the decision making process? What about the role of the overall health care system in the locations where women live? Do women have free access to preventative services and treatment in the event they are diagnosed with breast cancer? What about social disparities and their role in cancer inequalities as indicated in this region
Reviewer 4 Report
Comments and Suggestions for Authors
Title: The sociocultural influences on breast cancer screening among rural African women in South Africa
Thank you for the opportunity to review this paper which will be of interest to those in this field.
Abstract:
The term ‘rural African’ is used five times in the Abstract, which could suggest generalisation.
Line 13: ‘…among rural African women.’– Perhaps you could be specific – ‘among rural South African women.’
Line 14: Same observation as in line 13: ‘rural South African women’
Line 16: Same observation: rural South Africa – when you are referring to your study.
Introduction:
Line 64-65 ‘…breast cancer screening from a rural South African perspective where most Africa women present with…’ It may be better to say ‘where most women’ as you are referring to South Africa.
Theoretical Framework
Line 95: The ‘researcher’ should be the ‘researchers.’
Materials and Methods;
Line 103: You stated that this paper is the result of a ‘larger qualitative study’, in line 116 you stated ‘A purposive sampling method was used to recruit 22 participants for this study’ – not sure what you mean by ‘larger qualitative study’
Line 118: ‘researcher’ should be ‘researchers.’
Data collection
When and where did the interviews take place?
What language was used to conduct the interview?
What was the average duration of the interview?
It would also be good to give an example of the interview questions.
Line 137: researcher – perhaps you can say ‘one of the researchers’ (there are three authors in the paper).
Line 146 – ‘South African women.’
Results
Line 235 – ‘South African women.’
Line 333: You stated that ‘all participants agreed, however that raising awareness of breast cancer screening among family, friends, and communities would encourage more women to undergo screenings’. It would be good to use more quotes from the participants to support his claim.
Discussion:
Line 397: You stated that ‘traditionally, African women need the permission of their spouses before receiving medical care at a hospital or attending a health education forum’. This is the first time you are mentioning women need the permission of their spouses… Did this emerge from the present study?
Conclusions:
Line 407 Add ‘South’ Africa as you are reporting the current study.
The key findings need to be outlined in the conclusion.
Reviewer 5 Report
Comments and Suggestions for Authors
I congratulate the authors on their choice of research topic and methodology. I suggest the following improvements with regard to
METHODOLOGY
In the data collection section, mention whether the content outline and the interview script were used.
Please provide further information on the conduct of the interviews Were the interviews repeated? If so, how many? How long were the interviews or focus groups?
What were the criteria for selecting the women's treatment clinics? More information is needed about their location, funding, services or the population they serve.
It is not clear from the data analysis whether the themes were pre-identified or derived from the data.
DISCUSSION
I consider the discussion to be well written, coherent and consistent with the findings presented.
CONCLUSIONS
The main points are well written.
LIMITATIONS OF THE STUDY
I do not see any limitations of the study. I would be grateful if the authors could point them out.
Bibliographical REFERENCES
I noticed that the authors used more references that are more than 7 years old. Is it because there are no more up-to-date references in the literature? I think so. I suggest that you include more up-to-date references.
Round 2
Reviewer 3 Report
Comments and Suggestions for Authors
Much improved manuscript!